# Serological Response and Clinical Protection of Anti-SARS-CoV-2 Vaccination and the Role of Immunosuppressive Drugs in a Cohort of Kidney Transplant Patients

**DOI:** 10.3390/v14091951

**Published:** 2022-09-02

**Authors:** Pinchera Biagio, Carrano Rosa, Schiano Moriello Nicola, Salemi Fabrizio, Piccione Amerigo, Zumbo Giulia, Scotto Riccardo, Villari Riccardo, Romano Paolo, Spirito Lorenzo, Gentile Ivan

**Affiliations:** 1Section of Infectious Diseases, Department of Clinical Medicine and Surgery, University of Naples Federico II, 80131 Naples, Italy; 2Section of Nephrology, Department of Public Health, University of Naples Federico II, 80131 Naples, Italy; 3Section of Urology, Department of Neurosciences, Reproductive and Odontostomatological Sciences, University of Naples Federico II, 80131 Naples, Italy

**Keywords:** SARS-CoV-2, COVID-19, vaccination, transplant, immunosuppression, serological response

## Abstract

Vaccination against SARS-CoV2 represents a key weapon to prevent COVID-19, but lower response rates to vaccination have frequently been reported in solid organ transplant recipients. The aim of our study was to evaluate the rate of seroconversion to SARS-CoV-2 mRNA vaccines in a cohort of kidney transplant recipients and the potential role of the different immunosuppressive regimens. We conducted an observational retrospective cohort study in kidney transplant patients vaccinated for COVID-19. For each patient, we evaluated IgG anti-S-RBD SARS-CoV-2 titers immediately before the administration of first COVID-19 vaccination dose, 20 days after the first dose and 40 days after the second dose. Moreover, we evaluated the type of immunosuppressive treatment and the incidence of vaccine breakthrough SARS-CoV-2 infection. We enrolled 121 kidney transplant patients vaccinated for COVID-19. At the time of administration of the first vaccine dose, all patients had a negative antibody titer; only 4.1% had positive antibody titers 20 days after the first dose. More than half patients 62 (51%) had protective antibody titers 40 days after the second dose. A total of 18 Solid Organ Transplant Recipients (SOTRs) (14.9%) got a SARS-CoV-2 breakthrough infection during the study period. With regard to immunosuppressive regimen, patients on mycophenolate-based regimen (48.7%) showed the lowest antibody response rates (27.5%) compared to other regimens. Our study confirms that kidney transplant patients show a poor response to two doses of COVID-19 vaccination. Moreover, in our study the use of mycophenolate is significantly associated with a non-response to COVID-19 m-RNA vaccines.

## 1. Introduction

Vaccination against SARS-CoV-2 represents a key weapon to prevent COVID-19. Although efficacy of the available vaccines has been well established, lower response rates to vaccination have frequently been reported in immunosuppressed patients, such as solid organ transplant recipients (SOTRs) [1,2,3,4].

In fact, rates of seroconversion to vaccination have been shown to be low in immunocompromised patients. Moreover, seroconversion rate in SOTRs (37%) is even lower in comparison to patients with other conditions associated with immunosuppression, such as hematological malignancy (54.7%), solid tumors (82.4%), or HIV infection (94%) [5]. However, little is known about the mechanisms and molecular pathways involved in the very low rate of response in SOTRs [6].

Consistent with these observations, Sun et al. demonstrated that SOTRs have higher risk of breakthrough SARS-CoV-2 infection compared to people without immune dysfunction with an adjusted incidence rate ratio of 2.16 [95% CI: 1.96–2.38] [7].

The aim of our study was to evaluate the rate of seroconversion to SARS-CoV-2 mRNA vaccines in a cohort of kidney transplant recipients and the potential role of the different immunosuppressive regimens. Our secondary objective was to assess, in the same population, the incidence and outcome of breakthrough SARS-CoV-2 infections, and their association with post-vaccination antibody titers.

## 2. Materials and Methods

We conducted an observational retrospective cohort study in kidney transplant patients vaccinated for COVID-19 and on follow-up at “Federico II” University Hospital of Naples (Italy) from February 2021 to December 2021.

For each patient, we evaluated IgG anti-S-RBD SARS-CoV-2 titers (Roche Diagnostics GmbH, Mannheim, positive threshold >15 BAU/mL) immediately before the administration of first COVID-19 vaccination dose, 20 days after the first dose and 40 days after the second dose. Negative serology was defined, according to manufacturer, as an anti-S titer less than 15 BAU/mL.

Moreover, we evaluated for each patient main demographic characteristics as well as the type of immunosuppressive treatment at the time of vaccination as well as the incidence of vaccine breakthrough SARS-CoV-2 infection, defined as an RT-PCR-confirmed SARS-CoV2 infection with symptom onset or first positive RT-PCR rhino-oropharyngeal swab ≥14 days following the second dose of COVID-19 vaccine [8,9].

To describe the clinical status of patients with SARS-CoV-2 breakthrough infection we used the OSCI (Ordinal Scale for Clinical Improvement) scale [10]. The OSCI (Ordinal Scale for Clinical Improvement) is a 9-point scale, where 0 corresponds to no infection and 8 corresponds to death [10]. Patients with mild/moderate symptoms were defined as those presenting with at least one COVID-19 related symptom (e.g., cough, fever, sore throat, rhinorrhea), but with no need of oxygen therapy or hospitalization (OSCI Score: 0–2) [10].

The Kolmogorov-Smirnov test was applied to quantitative variables to check for Gaussian distribution. Quantitative data were reported as median and interquartile range (IQR) in case of non-parametric distribution. Data are given as mean ± standard deviation or as median and IQR in case of Gaussian and non-Gaussian distribution, respectively. For comparisons between continuous variables, the U Mann-Whitney Test was performed. We used the Chi-Square Test to test if two categorical variables are associated. Co-variates significantly associated with death at the univariate analysis were also analyzed in a multivariate model. The *p*-value for statistical significance was set at 0.05 for all the tests.

We apply Kaplan-Meier curves to show probability of survival in patients vaccinated with two doses over a six-month follow-up period.

With respect to the ethical issues, the study was led in accordance with ethical principles that have their origin in the Declaration of Helsinki and in good clinical practice. The study was reviewed and approved by the by the Institutional Review Board (or Ethics Committee) of Azienda Ospedaliera Universitaria Policlinico “Federico II”, Naples (protocol number 155/20). The authors confirm that the ethical policies of the journal have been observed.

## 3. Results

We enrolled 121 kidney transplant patients vaccinated for COVID-19 with a median age of 61 years (IQR, 25–88). Main demographic characteristics are summarized in Table 1.

All patients received two doses of mRNA BNT162b2 (Pfizer-BioNTech) COVID-19 vaccination, with a 3-week interval between doses.

At the time of administration of the first vaccine dose, all patients had a negative antibody titer; only 4.1% (*n* = 5) had positive antibody titers 20 days after the first dose. (Table 1) More than half patients 62 (51%) had protective antibody titers 40 days after the second dose.

All enrolled patients were on immunosuppressive therapy at the time of enrollment. Particularly, 62 patients (51.3%) were receiving a triple immunosuppressive therapy and 59 (48.7%) a dual therapy. (Table 1 and Table 2)

At the second pre-specified time-point (40 days after the second dose) roughly half patients had a negative serology (See Table 2). Notably, 28% of patients on dual immunosuppressive therapy did not develop antibody response after two doses of vaccine compared to 67% of patients undergoing triple immunosuppressive therapy (OR: 1.5 [95% CI: 0.35–0.92] triple vs. dual; *p*: 0.044) (Table 2).

A total of 18 SOTRs (14.9%) got a SARS-CoV-2 breakthrough infection during the study period (in the six months following the last vaccination dose). Of these, sixteen patients (88.8%) showed symptoms of the disease. In detail, 12 patients (75%) had a mild/moderate disease (OSCI score: 2–4) and 4 (25%) developed a severe disease (OSCI score: 5–6). Hospitalization was necessary for 6 (37.5%) patients. Notably, 12 (67%) patients who experienced a breakthrough infection showed no antibody response at the time of infection.

With regard to immunosuppressive regimen, patients on mycophenolate-based regimen (48.7%) showed the lowest antibody response rates (27.1%) compared to other regimens (*OR 1.4*, *95 CI* (*0.25–0.89*) mycophenolate-based vs. other non- mycophenolate-based regimens; *p*: *0.039*). (Table 2) The average age of patients with antibody response (27%) who were treated with mycophenolate based therapy was 47 (25–74).

No significant differences were observed regarding the risk of breakthrough infection (OR: 1.1, 95 CI (0.70–2.8); *p*: 0.430) and of developing COVID-19 (OR: 1.2, 95 CI (0.55–1.4); *p*: 0.280) between patients on mycophenolate-based regimen vs. patients treated with other mycophenolate.

No significant differences were observed among the different immunosuppressive drugs in the risk of vaccine breakthrough infection and in the evolution of COVID-19.

The Kaplan-Meier probabilities of death is shown in Figure 1. By 180 days, the probability of survival was about 93% in vaccinated with two doses.

## 4. Discussion

In our study only 51% of kidney transplant patients had a detectable antibody response after the second dose. This confirms what is already known in the literature [1]. In fact, our data are in line with those of by Marinaki S. et al. [11], who found a response rate to vaccination of 39.6% in a population of 455 SOTs vaccinated with COVID-19 mRNA SARS-CoV-2 vaccines and of 32.2% in kidney transplant recipients. Moreover, in a recently published systematic review including a total of 1744 SOT recipients [12], the authors reported a 33% of response to vaccine in kidney transplants recipients after a cycle of two doses [13]. The lower response rate found in kidney transplantation patients compared to other SOTs is probably attributable to the type of immunosuppression which is usually more intense than other SOTs [13].

In the study by Marinaki et al., younger age at transplantation, male gender, use of antimetabolite and steroid-free immunosuppression (IS) as well as the type of transplanted organ (heart and lung vs. kidney) were identified as the factors independently associated with better response to vaccine. [11]. In our study, age, gender, and steroid-free immunosuppression were not associated with the response to vaccination, whereas the use of mycophenolate was a significant risk factor for the lack of antibody production. Regarding the use of mycophenolate, our results agreed with what Manuel O. reported, while from the meta-analysis conducted by Anuraag Jena et al. the mycophenloate as a risk factor for a low response was not highlighted [14,15].

We acknowledge that our study has several limitations. One of the main limitations of our study is that we evaluated efficacy of only 2 doses of vaccine, which is currently considered a suboptimal protection compared to the 3 doses. Even in the setting of SOT, recent studies showed a higher rate of immune responders in 101 SOT recipients receiving a third dose of the BNT162b2 vaccine. In fact, they showed a raise in response rate from 40% after the second to 68% after administration of the third dose [16,17,18]. Another major limitation is the small sample size of our study which prevented to achieve definitive conclusions.

The major strength of our study is that it couples data on serology with rate of infection and clinical outcome of the patients.

Regarding the clinical relevance, it is known that vaccinated SOTs have a greater risk of vaccine breakthrough infections and a greater risk of developing COVID-19 after two doses of vaccination than the general population (15% vs. 8% and 11% vs. 5%) [17,18,19]. However, vaccinated SOTs have a lower risk of acquiring the infection and a lower risk of developing COVID-19 than unvaccinated SOTs (15% vs. 20–30% and 11% vs. 35–40%) [19,20,21]. In our study, we showed a similar incidence of breakthrough infections to data of the literature (14.6% vs. 15%).

It is noteworthy that the Kaplan-Meier curves showed that there was a significantly higher likelihood of SARS-CoV-2 infection in patients with negative serology than in patients with positive serology (Figure 1), while no difference was observed in relation to symptomatic infections (COVID-19) or severe forms of COVID-19 for the two patient groups (negative vs. positive serology) during the 210-day follow-up period (Figure 2 and Figure 3). However, we acknowledge that the data regarding symptomatic and severe diseases should take into account the small number of events reported.

## 5. Conclusions

In conclusion, our study confirms that kidney transplant patients show a poor response to two doses of COVID-19 vaccination. Therefore, these results should prompt these patients to perform additional booster doses and also to explore different prevention strategies such as passive prophylaxis with monoclonal antibodies. Moreover, in our study the use of mycophenolate is significantly associated with a non-response to COVID-19 m-RNA vaccines. Further studies are needed to confirm these data.

## Figures and Tables

**Figure 1 viruses-14-01951-f001:**
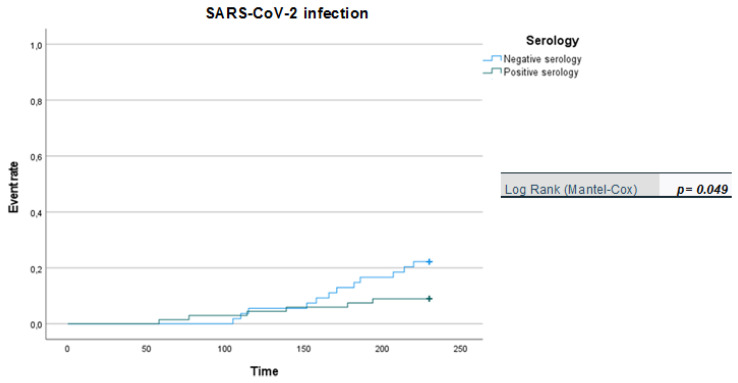
Kaplan-Meier curves: positive vs. negative serology in breakthrough infection (in days).

**Figure 2 viruses-14-01951-f002:**
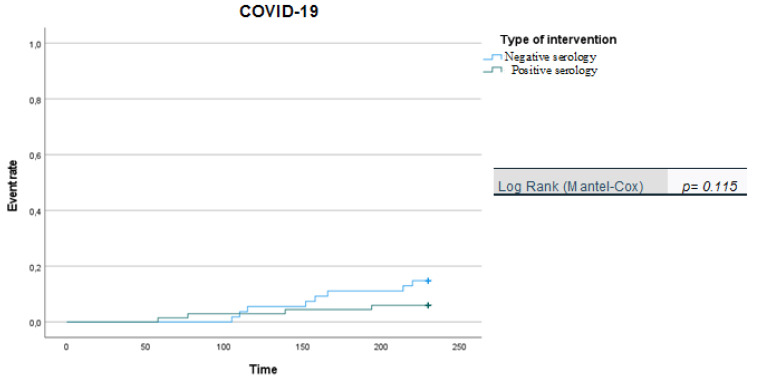
Kaplan-Meier curves: positive vs. negative serology in COVID-19 (in days).

**Figure 3 viruses-14-01951-f003:**
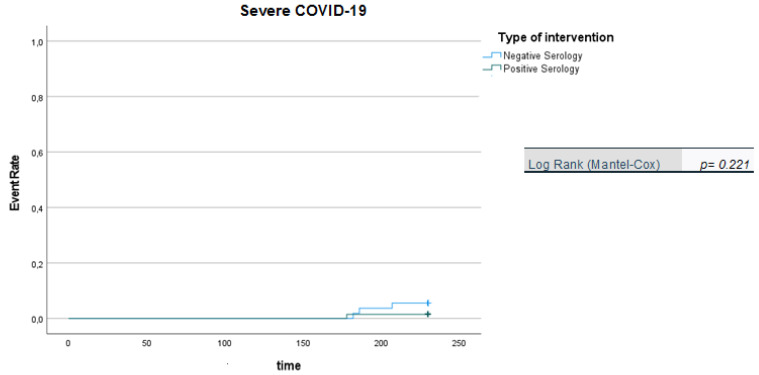
Kaplan-Meier curves: positive vs. negative serology in Severe COVID-19 (in days).

**Table 1 viruses-14-01951-t001:** Characteristics of enrolled patients (N = 121).

** *Age (Median, IQR)* **	** *61 (25–88)* **
** *Gender* ** *M* *F*	*74 (61%)* *47 (39%)*
** *Age ≤ 25 (%)* **	*5 (4%)*
** *Gender* ** ** *M* ** ** *F* **	*4 (80%)* *1 (20%)*
** *Time period between transplantation and COVID-19 vaccination (months), median (IQR)* **	*108 (12–384)*
** *Duration of post-vaccination follow-up (months), median (IQR)* **	*5 (3–9)*
** *Immunosuppressive therapy at the time of enrollment* ** *Tacrolimus-Mycophenolate-Steroids* *Tacrolimus-Everolimus-Steroids* *Cyclosporine-Mycophenolate-Steroids* *Other immunosuppressive therapies*	*35 (29%)* *10 (8%)* *9 (7.5%)* *67 (55.5%)*
** *Incidence SARS-CoV-2 breakthrough Infection* **	*18 (14.9%)*
** *Ig anti-SARS-CoV-2 titer (BAU/mL)* ** ** *Pre-vaccination* ** ** *Positive* ** ** *20 days after the first dose* ** ** *Positive* ** ** *40 days after the second dose* **	*0* *5 (4.1%)* *62 (51%)*

**Table 2 viruses-14-01951-t002:** Immunosuppressive therapy and Ig anti-SARS-CoV-2 post-vaccination serological status (N = 121).

Immunosuppressive Therapy			
		Seropositivity	*p-Value*
**Double**	59 (48.7%)	42 (71%)	*0.044*
**Triple**	62 (51.3%)	20 (32%)	
**Tacrolimus-** **containing regimens**	90 (74.3%)	44 (49%)	*0.275 **
**Mycophenolate-containing regimens**	59 (48.7%)	16 (27%)	*0.039 **
**Steroid-** **containing regimens**	105 (86.8%)	56 (53%)	*0.450 **
**Everolimus-** **containing regimens**	17 (14%)	9 (53%)	*0.530 **
**Cyclosporine-containing regimens**	23 (19%)	12 (52%)	*0.285 **
**Sirolimus-** **containing regimens**	7 (5.8%)	6 (86%)	*0.320 **
**Azathioprine-** **containig regimens**	3 (2.5%)	1 (33%)	*0.254 **

* vs. other treatments.

## Data Availability

It is possible to request the data from the corresponding author.

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
