# Peer review of "Serological Response and Clinical Protection of Anti-SARS-CoV-2 Vaccination and the Role of Immunosuppressive Drugs in a Cohort of Kidney Transplant Patients"

_viruses, 2022, doi:10.3390/v14091951_

Round 1

Reviewer 1 Report

In this study, the authors intend to evaluate the rate of seroconversion to SARS CoV-2 mRNA vaccines in a cohort of kidney transplant recipients and the potential role of the different immunosuppressive regimens. Their study confirms that kidney transplant patients show a poor response to two doses of COVID-19 vaccination. Moreover, the use of mycophenolate is significantly associated with a non-response to COVID-19 mRNA vaccines.

Several suggestions:

1.      Use [SARS-CoV-2] but not other typos, e.g., SARS-CoV 2 (line 1). Please check the entire manuscript.

2.      Lines 31 and 107, 62 (51%) but not (62.51%).

3.      Line 83, please delete [interquartile range].

4.      Table 1, 47(49%) should be [47(39%)].

5.      Regarding Table 2, please explain what is [*].

6.      Line 128, how to get the 27.5%? [16/59=27.1%]

7.      Line 149, is there an [patients] after [kidney transplantation]?

Author Response

Response to Reviewer 1:

I thank the Reviewer for the revisions, advice and suggestions, I believe that the revisions made have contributed significantly to the improvement of the manuscript.

Response to several suggestions:

  1. Use [SARS-CoV-2] but not other typos, e.g., SARS-CoV 2 (line 1). Please check the entire manuscript.

          I proceeded to carry out this revision, as tracked in the manuscript.

  1. Lines 31 and 107, 62 (51%) but not (62.51%).

          I proceeded to carry out this revision, as tracked in the manuscript.

  1. Line 83, please delete [interquartile range].

          I proceeded to carry out this revision, as tracked in the manuscript.

  1. Table 1, 47(49%) should be [47(39%)].

          I proceeded to carry out this revision, as tracked in the manuscript.

  1. Regarding Table 2, please explain what is [*].

          I proceeded to carry out this revision, as tracked in the manuscript.

  1. Line 128, how to get the 27.5%? [16/59=27.1%]

          I proceeded to carry out this revision, as tracked in the manuscript.

  1. Line 149, is there an [patients] after [kidney transplantation]?

         I proceeded to carry out this revision, as tracked in the manuscript.

Reviewer 2 Report

This article has been presented in a nice way. The article explains the importance of vaccination in patient with Solid Organ Transplant since they go through immunosuppressive therapy and it could interfere with the vaccine responses to the virus. I have few points to add that would improve the clarity of the paper.

1. In Table 1., the author needs to include the number of patients included in study that are close to 25 years of age in male and female group.

2. Table 1, all the texts must be aligned properly to understand the data.

3. The author needs to include the average age of patients (27%) who were treated with mycophenolate based therapy.

4. Figure 1, Figure 2 and Figure 3., the Y axis level can be reduced to 0.4 which would help readers see the difference in event rate for Negative and Positive serology groups.

5. X axis is time in days?

Author Response

Response to Reviewer 2:

I thank the Reviewer for the revisions, advice and suggestions, I believe that the revisions made have contributed significantly to the improvement of the manuscript.

Response to several suggestions:

  1. In Table 1., the author needs to include the number of patients included in study that are close to 25 years of age in male and female group.

I have reported in table 1 what is requested, as outlined in the manuscript, as tracked in the manuscript.

  1. Table 1, all the texts must be aligned properly to understand the data.

    I proceeded to carry out this revision, as tracked in the manuscript.

  1. The author needs to include the average age of patients (27%) who were treated with mycophenolate based therapy.

As requested, I proceeded to insert this data in the text, as tracked in the manuscript.

  1. Figure 1, Figure 2 and Figure 3., the Y axis level can be reduced to 0.4 which would help readers see the difference in event rate for Negative and Positive serology groups.

I proceeded to modify the figures.

  1. X axis is time in days?

Yes, the time is in days. As suggested, I have corrected and specified in the text the time in days, as tracked in the mauscript.

Reviewer 3 Report

In this paper, the authors have analysed the serological response afetr vaccination against SARS-CoV-2 in kidney transplanted patients. They observe a poor response to 2 doses of vaccine and particularly in patients treated by mycophenolate. Tis last aspect should be better discussed and some references are missing, i. e.  the prospective study  Clin Microbiol Infect. 2021 Aug;27(8):1173.e1-1173.e indiciating results similar to yours, or a meta-analysis published in Autoimmun Rev. 2022 Jan;21(1):102927. doi: 10.1016/j.autrev.2021.102927, which did not identify mycophenloate as a risk factor for a low response (but it was not focused only on kydney transplantation).

Minor comments: 

- Abbreviation "SOTR" in the abstract is not explained.

- The layout of table 2 needs to be corrected

Author Response

Response to Reviewer 3:

I thank the Reviewer for the revisions, advice and suggestions, I believe that the revisions made have contributed significantly to the improvement of the manuscript.

Response to several suggestions:

  • “In this paper, the authors have analysed the serological response after vaccination against SARS-CoV-2 in kidney transplanted patients. They observe a poor response to 2 doses of vaccine and particularly in patients treated by mycophenolate. This last aspect should be better discussed and some references are missing, i. e. the prospective study  Clin Microbiol Infect. 2021 Aug;27(8):1173.e1-1173.e indiciating results similar to yours, or a meta-analysis published in Autoimmun Rev. 2022 Jan;21(1):102927. doi: 10.1016/j.autrev.2021.102927, which did not identify mycophenloate as a risk factor for a low response (but it was not focused only on kydney transplantation)”.

As suggested, I have expanded the discussion and cited the two indicated manuscripts (Manuel O. COVID-19 vaccination in solid-organ transplant recipients: generating new data as fast as possible, but taking clinical decisions as slow as necessary. Clin Microbiol Infect. 2021 Aug;27(8):1070-1071, Anuraag Jena, Shubhra Mishra, Parakkal Deepak et al. Response to SARS-CoV-2 vaccination in immune mediated inflammatory diseases: Systematic review and meta-analysis.  Autoimmun Rev . 2022 Jan;21(1):102927), as tracked in the manuscript.

  • Abbreviation "SOTR" in the abstract is not explained.

   I proceeded to carry out this revision, as tracked in the manuscript.

  • The layout of table 2 needs to be corrected.

   I proceeded to carry out this revision, as tracked in the manuscript.
